# High Sensitivity Temperature Sensing of Long-Period Fiber Grating for the Ocean

**DOI:** 10.3390/s23104768

**Published:** 2023-05-15

**Authors:** Jiayi Qu, Hongxia Zhang, Xinyu Shi, Chuanxi Li, Dagong Jia, Tiegen Liu, Rongxin Su

**Affiliations:** 1Key Laboratory of Optoelectronics Information Technical Science, College of Precision Instrument and Opto-Electronics Engineering, Tianjin University, Tianjin 300072, China; jiayiqu_00@163.com (J.Q.); shixinyu960422@163.com (X.S.); dagongjia@tju.edu.cn (D.J.); tgliu@tju.edu.cn (T.L.); 2Changchun National Extreme Precision Optics Co., Ltd., Changchun 130033, China; 3State Key Laboratory of Chemical Engineering, School of Chemical Engineering and Technology, Tianjin University, Tianjin 300072, China; 4Tianjin Key Laboratory of Membrane Science and Desalination Technology, School of Chemical Engineering and Technology, Tianjin University, Tianjin 300072, China; 5School of Marine Science and Technology, Tianjin University, Tianjin 300072, China

**Keywords:** ocean temperature sensing, long-period fiber grating, coupling mode conversion, magnetron sputtering coating, package temperature sensitivity enhancement

## Abstract

In this study, a new temperature sensor with high sensitivity was achieved by four-layer Ge and B co-doped long-period fiber grating (LPFG) based on the mode coupling principle. By analyzing the mode conversion, the influence of the surrounding refractive index (SRI), the thickness and the refractive index of the film on the sensitivity of the sensor is studied. When 10 nm-thick titanium dioxide (TiO_2_) film is coated on the surface of the bare LPFG, the refractive index sensitivity of the sensor can be initially improved. Packaging PC452 UV-curable adhesive with a high-thermoluminescence coefficient for temperature sensitization can realize high-sensitivity temperature sensing and meet the requirements of ocean temperature detection. Finally, the effects of salt and protein attachment on the sensitivity are analyzed, which provides a reference for the subsequent application. The sensitivity of 3.8 nm/°C in the range of 5–30 °C was achieved for this new sensor, and the resolution is about 0.00026 °C, which is over 20 times higher than ordinary temperature sensors. This new sensor meets the accuracy and range of general ocean temperature measurements and could be used in various marine monitoring and environmental protection applications.

## 1. Introduction

The ocean is an important strategic space to ensure sustainable development. Seawater temperature is one of the significant parameters in the marine environment; detecting and mastering the real-time change and space–time distribution is a major content of marine observation [1]. Global ocean temperature usually varies between −2 and 30 °C, and more than 50% of the oceans have an average seawater temperature higher than 20 °C annually. The ocean temperature in the deep sea is relatively stable, and the average daily variation is about 0.001 °C. In order to detect the small changes in seawater temperature, the measurement accuracy and temperature range of current frontier research is required to be ±0.001 °C and about 5–30 °C.

At present, the demand for ocean temperature sensors is mainly in miniaturization, high accuracy, large range and high stability. Traditional electrical temperature detectors are expensive, and it is difficult to establish a wide-coverage submarine observation network. A fiber grating sensor uses an optical fiber as the medium to sense and transmit external signals through light. The measurement method mainly depends on the wavelength demodulation. The interference of optical fiber bending, connection loss, light source fluctuation and other problems is small [2]. After calibration, an absolute measurement can be realized, which is convenient for reuse and array measurements. Fiber grating sensors have the characteristics of small size, electrical insulation, high sensitivity and electrical insulation, and have been extensively used in displacement sensing, strain sensing, temperature sensing, refraction measurement sensing, and other fields [3]. According to the sensing principle, the fiber optic technology of temperature sensing mainly includes fiber grating (FG), interferometer optic fiber systems (OFS) [4,5,6], photogenic crystal fiber (PCF)-based [7], surface plasmon resonance (SPR) [8], optical microfiber coupler (OMC) [9,10], etc. Interferometer sensors include the Fabry–Pérot interferometer (FPI) [4], the Mach–Zehnder interferometer (MZI) [5], the Sagnac interferometer (SI) [6], etc. Most interferometer OFS can only be used for single- or dual-parameter sensing, and there are some problems, such as limited sensitivity and poor structural stability. For PCF sensors, highly sensitive temperature sensing can be achieved by filling the holes of the PCF with specific refractive index-matching fluid (IMF). SPR sensors generally have high sensitivity and can meet the requirements of high-precision testing in the marine environment. The performance of OMC sensors can be comparable to commercial electrical sensors. However, the key problems of fiber optic sensors, such as environmental adaptability and multi-parameter cross-sensitive demodulation, need to be further studied [11]. Fiber grating is classified into fiber Bragg grating (FBG) and long-period fiber grating (LPFG) according to the size of the grating period. The temperature sensitivity of ordinary FBG and LPG is about 10 pm/°C and 100 pm/°C, respectively [12,13]. In order to achieve the measurement requirement of ±0.001 °C, the temperature sensing sensitivity should be more than 1 nm/°C when using commonly used fiber grating demodulators with an accuracy of 1 pm. Therefore, it is necessary to further increase the temperature sensitivity of fiber grating sensors [14] by comparing the basic performance of sensors. Although FG sensors have relatively low sensitivity, their strong stability and environmental adaptability give them strong application value [15].

In recent years, with the rapid development of thin film technology, the combination of fiber grating technology with materials, chemistry, biology and other technologies has opened up many new applications of LPFG. By coating specific materials (such as metal, polymers, etc.) on fiber gratings, multi-parameter sensing can be realized, and the problem of cross-sensitivity of multi-parameters can be solved [15]. By coating nanofilm with a high refractive index on the fiber surface, the refractive index sensitization can be effectively realized. Titanium dioxide (TiO_2_) is widely used in the optical and biological fields due to its high refractive index, non-toxicity and high biocompatibility. Sensors made of fiber grating sputtered with TiO_2_ have been widely used in refractive index measurements [16,17]. In order to obtain higher sensitivity, thermosensitive materials with a high thermoluminescence coefficient and thermal expansion coefficient have attracted increasing attention from researchers. These have been successfully applied to improve the response of optical fiber sensors to temperature [13]. Packaging a high refractive index layer is conducive to the conversion between the cladding guide mode and the coating guide mode, avoiding the problem of cross-sensitivity between the temperature and SRI [12]. Recently, temperature measurement has been mainly realized by packaging with polyvinyl alcohol (PVA) [18], polyacrylate (PAA) [12,19], urethane acrylate [20], polydimethylsiloxane (PDMS) [13,21,22] and polymethylmethacrylate (PMMA) [23,24,25], but the temperature test ranges still cannot meet the needs of marine measurement.

In this study, a new sensor with high sensitivity was achieved by using a special long-period fiber grating doped with Ge and B. According to the mode coupling theory, by sputtering TiO_2_ nanofilm on the side of LPFG, we can achieve a preliminary refractive index sensitization. Then, by packaging the material with a high thermoluminescence coefficient, the refractive index sensitivity is converted into temperature sensitivity by using the change of the refractive index of the packaging material with temperature. Because the sensor needs to be immersed in the ocean for a long time in practical applications, we test the impact of seawater salinity and marine biological protein adhesion on the sensor. Through the surface modification of the chimeric polymer coating, the anti-protein adhesion ability of the sensor is improved, which provides a reference for improving its stability in the marine environment. When the 10 nm-thick TiO_2_ film is sputtered on the surface of Ge and B co-doped LPFG and packaged with PC452 UV-curable adhesive, the maximum temperature measurement range is 5–30 °C, and the maximum temperature sensitivity and temperature resolution would be 3.8 nm/°C and 0.00026 °C if the resolution of the optical spectrum analyzer was 1 pm, which meets the test requirements of ocean temperature sensing.

## 2. Theory and Simulation

Modes can characterize the distribution of light waves in space. When the fiber is modulated, such as when the refractive index or structure size changes, the modes in the core and cladding undergo energy transfer, which is called mode coupling. LPFG belongs to the transmissive fibers, and its internal mode coupling comes from the energy transfer of the basic mode and cladding mode, both of which propagate along the forward direction. When LPFG is fabricated on a single-mode fiber, its periodic modulation changes the dielectric constant and the orthogonal independent of the mode of the fiber. Its refractive index variation is usually less than 10^−2^, and the existence of the gate region will not have a significant impact on the distribution of the eigenmode field. Each coupling mode can be regarded as a linear superposition of different eigenmodes, so we only need to consider the energy transfer between the different modes.

### 2.1. Theory of LPFG Refractive Index Sensing

The four-layer cylindrical waveguide model of the fiber core–cladding–nanofilm external environment is shown in Figure 1b. The refractive index of the core material, cladding material, film material and SRI are set to $n_{1}$, $n_{2}$, $n_{3}$ and $n_{SUR}$, respectively.

According to the mode coupling theory, the resonant peak wavelength, $\lambda_{D}$, of an ideal LPGF is:(1)$$\lambda_{D}=\left(n_{1eff}-n_{2eff}\right)\cdot \Lambda$$
where $n_{1eff}$ and $n_{2eff}$ are the effective refractive indices of the core mode and cladding modes, and Λ is the period of the LPFG. The starting point of the LPFG modulation range is regarded as the origin, and the energy value of the LPFG fundamental mode at the origin is regarded as 1 ($\left|A^{2}\left(0\right)\right|=1$). When the light wave passes through the grid region with a distance of *L*, the energy in the fundamental mode will be transferred to the cladding mode. The residual energy in the fundamental mode at the end of the gate area is $\left|A^{2}\left(L\right)\right|$, and the expression of LPFG transmissivity is derived:(2)$$T\left(\lambda \right)={\left|A\left(L\right)\right|}^{2}=\cos^{2}\left(\sqrt{{\hat{\sigma }}^{2}+\kappa^{2}}L\right)+\frac{1}{1+\kappa^{2}/{\hat{\sigma }}^{2}}\sin^{2}\left(\sqrt{{\hat{\sigma }}^{2}+\kappa^{2}}L\right)$$
wherein κ and $\hat{\sigma }$ are the coupling coefficients of “AC” and “DC”. The above formula can be used to calculate the transmission spectrum generated when the basic mode in the core of the LPFG is coupled with a mode in the cladding. The complete transmission spectrum of the LPFG can be obtained by superposition of the transmission spectra of all the modes in the cladding. When $n_{SUR}$ changes, the energy in the LPFG core will be coupled to the cladding, and the $\lambda_{D}$ of the transmission spectrum will shift because of the influence of the evanescent field. Due to the low refractive index sensitivity and limited measurement range of ordinary LPFG, we can greatly improve the SRI sensitivity of LPFG by coating nanofilms on the surface of LPFG with high refractive index materials that are much higher than that of optic fiber’s cladding [12].

The effective refractive index of the cladding mode $n_{2eff}$ is modulated by $n_{3}$, film thickness and $n_{SUR}$ at the same time. When $n_{SUR}$ is raised to a certain threshold, the optical field in the cladding undergoes mode conversion. The range of $n_{SUR}$ that causes the mode conversion is called the conversion region. According to Formula (1), in the conversion region, $n_{2eff}$ changes rapidly and the $\lambda_{D}$ also has a large drift, so it has a high refractive index sensitivity. The mode conversion phenomenon of LPFG can provide a reference for the enhancement of refractive index and temperature sensitivity.

### 2.2. Simulation of Mode Coupling Theory of Coated LPFG

We established a four-layer model in OptiGrating software to simulate the transmission spectrum and resonance peak offset of LPFG, as shown in Figure 1. The simulation data of ordinary single-mode optical fibers and special single-mode optical fibers co-doped with Ge and B are as Table 1.

The parameter of grating is set to the period length of 400 μm, the number of periods of 60, $n_{3}$ = 2.163, and the film thickness of 15 nm, 25 nm and 50 nm, respectively. In the range of $n_{SUR}$ from 1.33 to 1.146, the response of the $n_{2eff}$ of each cladding mode to $n_{SUR}$ at the simulated wavelength of 1550 nm is shown in Figure 2.

When the coating thickness is 10 nm, the $\lambda_{D}$ offset in the LP (0.5)–LP (0.8) mode with the SRI is shown in Figure 3a. It is easy to see that the sensitivity of the $\lambda_{D}$ of the LP (0.5) mode is significantly lower than that of the LP (0.8) mode. The sensitivity of the refractive index sensing increases as the mode order increases. Therefore, within the measurement and demodulation range of the sensing system, the better refractive index sensing sensitivity can be obtained by selecting the $\lambda_{D}$ of the mode with the higher mode for analysis.

According to the analysis in 2.1, when $n_{SUR}$ approaches $n_{2}$ (1.45–1.46), mode conversion occurs. With the sample coated film with the 10 nm thickness, the mode conversion range of LP (0.5) and LP (0.6) modes is 1.44–1.45, which is closer to $n_{2}$ than the other modes. Because the refractive index sensitivity of the higher mode is better, LP (0.6) is chosen for the simulation and comparison.

Figure 3b compares the relationship between the $n_{2eff}$ of the LP (0.6) mode and $n_{SUR}$. It can be seen from Figure 2 and Figure 3b that, with the increase in the film thickness, the slope of the $n_{2eff}$–$n_{SUR}$ curve becomes slower. The range of the mode conversion region becomes larger and moves to the shortwave direction. With the same film thickness, the $n_{2eff}$–$n_{SUR}$ curve slope of higher-order modes in the conversion region is relatively larger. This means that the corresponding resonant peak has higher sensing sensitivity.

Figure 4 shows the response of the $n_{2eff}$ of the LP (0.6)–LP (0.9) mode when the coating thickness is 50 nm, $n_{SUR}$ = 1.0, and $n_{3}$ changes from 1.5 to 3.5. When $n_{3}$ is in the range of 2.1–2.4, the mode is converted to a higher order. In the conversion region, the $n_{2eff}$ and $\lambda_{D}$ shift sharply, so $n_{3}$ also has a great impact on the mode coupling of LPFG.

When setting $n_{3}$ = 2.163 and $n_{SUR}$ = 1.0, we simulated the change of the effective refractive index and $\lambda_{D}$ of the LP (0.6)–LP (0.9) mode in the range of coating thickness from 0–700 nm, as shown in Figure 5. When the film thickness changes from 60 nm to 120 nm, the cladding modes convert to the higher modes in turn. Continuing to increase the coating thickness shows that the mode conversion phenomenon is periodic.

### 2.3. Theory of LPFG Temperature Sensing

Using Formula (1), we can derive the analytic expression for the temperature sensitivity of the resonant wavelength as [26]:(3)$$\frac{d\lambda_{D}}{dT}=\lambda_{D}\cdot \gamma \cdot \left(\alpha +\Gamma_{temp}\right)$$
where *α* is the coefficient of thermal expansion; and *γ* describes the waveguide dispersion and is defined by [27]:(4)$$\gamma =\frac{\frac{d\lambda_{D}}{d\Lambda }}{n_{1eff}-n_{2eff}^{m}}$$

γ is also known as the general sensitivity factor. $\Gamma_{temp}$ describes the temperature dependence of the waveguide dispersion and is defined by:(5)$$\Gamma_{temp}=\frac{\xi_{1}n_{1eff}-\xi_{2}n_{2eff}^{m}}{n_{1eff}-n_{2eff}^{m}}$$
where $\xi_{1}$ and $\xi_{2}$ are the thermoluminescence coefficients of the core and cladding materials, respectively. Due to the influence of α and ξ, when the surrounding temperature changes, $n_{1eff}$, $n_{2eff}$ and the parameters of LPFG will be affected, which causes the shift of the resonant peak. Concerning the different order of the coupling mode, the temperature sensitivity also varies greatly.

Figure 6 shows the simulation of the temperature sensitivity of cladding mode LP (0.6) in two kinds of LPFG (the grating period is 400 μm; the number of periods is 60). When the initial properties of LPFG (Λ, refractive index modulation distribution, number of coupled modes m) are consistent, with the increase in the surrounding temperature, the $\lambda_{D}$ offset of these two kinds of LPFG will be exactly reverse. The response of ordinary single-mode LPFG is the red shift, with a sensitivity of about 0.1125 nm/°C, while the $\lambda_{D}$ of the Ge and B co-doped LPFG will shift blue, with a sensitivity of about −0.659 nm/°C, which is more than five times the ordinary single-mode LPFG.

However, the temperature sensitivity of unpackaged samples needs to be enhanced. Due to coated LPFG having high refractive index sensitivity, the mechanism of high thermoluminescence coefficient packaged LPFG is to transfer the high refractive index sensitivity to high temperature sensitivity and avoid the problem of cross-sensitivity between the temperature and $n_{SUR}$ [19,23]. Because the diameter of the LPFG is small, the refractive index of the packaging material can be seen as $n_{SUR}$. When the temperature changes, the expression of the relative change of $\lambda_{D}$ is:(6)$$\frac{\Delta \lambda_{D}}{\lambda_{D}}=K_{T}\Delta T+K_{\varepsilon }\left(\alpha_{1}-\alpha_{2}\right)\Delta T+\frac{\Delta \lambda_{SUR}}{\lambda_{D}}$$
where $K_{T}$ is the temperature response factor of LPFG; $K_{\varepsilon }$ is the force response factor; $\alpha_{1}$ and $\alpha_{2}$ are the thermal expansion coefficients of the packaging and the LPFG itself. $\Delta \lambda_{SUR}$ characterizes the position shift caused by the packaging layer (SRI) in response to the ambient temperature [26]:(7)$$\Delta \lambda_{SUR}=-\frac{u_{m}^{2}\lambda_{D}^{3}n_{SUR}}{8\pi r_{cl}^{3}n_{2}\left(n_{1eff}-n_{2eff}^{m}\right){\left(n_{2}^{2}-n_{SUR}^{2}\right)}^{3/2}}$$
where $u_{m}$ is the m-th root of the zeroth-order Bessel function of the first kind; $r_{cl}$ is the radius of the fiber cladding. It can be seen from Formula (7) that when the surrounding temperature changes, there are three main factors for the offset of the $\lambda_{D}$ of the PLPFG:(1)The thermal expansion and thermoluminescence coefficient of the LPFG’s materials;(2)The thermal expansion of the packaging material;(3)The thermoluminescence coefficient of the packaging material.

### 2.4. Simulation of Temperature Sensing Response Time

The thermodynamic analysis of the sensor structure is carried out by finite element analysis (FEA). The FEA model of the coated LPFG is established, as shown in Figure 7a. The film thickness is much lower than the UV-curing adhesive packaging. The thermal conductivity of the core and the cladding are almost the same, and the coated LPFG can be seen as a single material cylinder. The geometric parameters of the model are set as follows: the diameter of the coated LPFG is 125 μm, and the diameter of the sensor is 2 mm. The tetrahedral division method is selected for the grid division. The maximum side length of all the grids is selected to be 0.1 mm, and the side length of the local grids in the coating grating part is set to be 0.02 mm for the grid refinement. With the thermal conductivity of the LPFG material set to 1.46 W/(m·K), the specific heat is 722 J/(kg·K), the thermal conductivity of the packaging material is 0.4 W/(m·K), and the specific heat of the coating material is 1740 J/(kg·K). The temperature load is added to the side of the cylindrical model as the surrounding temperature changes, and the convection boundary condition is added to the end-face of the model. The default value of the convective heat transfer coefficient in a uniform water environment (1.2 × 10^−3^ W/(mm^2^/°C)) was selected.

In this paper, the temperature response time of the sensor is defined as the time that the temperature of fiber core takes to change by 90% of the difference between the surrounding temperature and the initial temperature when the surrounding temperature changes step-by-step. The response time is linked to the thickness and thermal conductivity of the packaging material. Figure 7b shows when the initial temperature of the sensors is set to 20 °C, the surrounding temperature is set to 30 °C, the temperature change curve of the fiber core and the cross-section temperature distribution of the sensor when the core temperature reaches 29 °C.

## 3. Experimental Section

### 3.1. Reagents and Materials

The chemical reagents required for this experiment include two kinds of UV-curing adhesives (PC452, Streamway International Trading (Shanghai) Co., Ltd., Shanghai, China; NOA 144, Norland Products Inc., Jamesburg, NJ, USA), a protein solution, a large amount of deionized water and glycerol. All the chemical reagents were used without further purification. Table 2 shows the relevant parameters of the two UV-curing adhesives. The optical fibers required for this experiment include a common single-mode fiber and a Ge and B co-doped photosensitive fiber (PS 1250/1500).

We prepared 14 standard refractive index samples with different refractive indices from 1.333 to 1.474 with a glycerol–deionized water mixed solution. The refractive index of each sample is measured with an Abbe refractometer, as shown in Table 3.

### 3.2. Preparation of the LPFG

The LPFG was fabricated by using a CO_2_ laser marking machine (CO2-G1, Han’s Laser Technology Industry Group Co., Ltd., Shenzhen, China) with the line-by-line method. The fabricating device system includes the control software, the CO_2_ laser, an optical fiber clamping device and a 3D displacement adjustment platform. Figure 8a shows the experimental setup of the LPFG fabricating process. The setting parameters are as follows: pulse frequency 5 kHz; wavelength 10.64 μm; max marking power 10 W; and marking position accuracy ±0.06 μm. The obtained LPFG samples have a grating period of 400 μm, the number of periods is 60, and the total length of the grating area is 2.4 cm.

Before fabricating, clamp the optical fiber to the optical fiber holder. Calibrate by fine-tuning the 3D displacement adjustment platform to keep the optical fiber axis horizontal and strictly parallel to the focal spot of the outgoing laser. During fabrication, make the beam mark transfer along the axis direction on the fiber to form the grating area. Use a real-time detection system composed of a broadband light source, fiber grating and spectrometer to measure the resonant peak wavelength and depth of the LPFG. Figure 8b shows an optical microscope photo of the fiber after fabricating. Figure 8c shows the transmission spectrum of the LPFG.

### 3.3. Coating and Packaging

In this work, a magnetron sputtering system (LN-CK4, Shenyang Lining Vacuum Technology Research Institute Co., Ltd., Shenyang, China) and a TiO_2_ target (purity 99.99%, diameter 80 mm, thickness 3 mm, and 2 mm-thick Cu back target is bound, Shijiazhuang Teke New Material Technology Co., Ltd., Shijiazhuang, China) were used for coating. The optical fiber clamping rotary table is used for clamping so that the film is evenly sputtered on the sample surface. A remote-control FET module is used to control the motor rotation, and the motor drives the fiber grating clamped in the gear shaft to rotate. TiO_2_ films with a thickness of about 10 nm and 15 nm were obtained by coating for 10 min and 15 min. The refractive index and thickness of the TiO_2_ films were measured with an ellipsometer (RC-2, J.A. Wollam Co., Ltd., Lincoln, NE, USA). The refractive index corresponding to the 1550 nm wavelength is 2.163483, which is consistent with the parameters selected in the simulation. Figure 9 shows the SEM diagram of cross-section of coated LPFG sample.

Two kinds of UV-curable adhesives (PC452, Streamway International Trading (Shanghai) Co., Ltd., Shanghai, China; NOA 144, Norland Products Inc., Jamesburg, NJ, USA) were used for packaging. A capillary tube with a length of about 3 cm and about 2.5 mm in diameter is sheathed in the grating area. The samples are clamped to the optical fiber holder to keep the grating area horizontally tensioned. Then, the adhesive is dropped at one side of the sample until the capillary tube is finally filled with reagent. During the curing process, the refractive index of the UV-curable adhesive will cause the offset of the resonance peak. Therefore, a supercontinuum spectral light source and spectrometer are used for real-time monitoring. When continuous irradiation no longer affects the resonant peak wavelength, the UV lamp can be turned off to end the curing. The object diagram of the sensor after packaging is shown in Figure 10.

### 3.4. Instrumentation

Figure 11 presents the schematic and physical picture of the experimental system. The excitation light was provided by a supercontinuum spectral light source (SC-54, Wuhan Yangtze Soton Laser Co., Ltd., Wuhan, China), and the transmission spectrum was measured using an AQ 6370C Telecom Optical Spectrum Analyzer (600–1700 nm, YOKOGAWA China Co., Ltd., Shanghai, China). A single-mode optical fiber mounting frame (APFP-FH, Beijing Zhuolihan Light Co., Ltd., Beijing, China) is used to fix the grating area when testing the samples, and the gate region is fixed in a horizontal tensional state to avoid cross-sensitivity of strain. During the test, the semiconductor temperature console, whose accuracy is 0.1 °C, is used to control the test temperature. The test temperature range is −5–30 °C.

## 4. Results and Discussion

### 4.1. Measurement of Refractive Index Sensitivity of LPFG

We measured the refractive index sensitivity of uncoated LPFG in the range of 1.33–1.47. Figure 12a,b shows the comparison of the simulation and experimental results of two kinds of LPFG. The experimental results fit well with the simulation results.

Figure 13a shows the relationship between the transmission peaks of the LP (0.6) mode and $n_{SUR}$. When the $n_{SUR}$ is close to 1.46 ($n_{2}$), the transmission peak shifts blue and the transmission depth decrease rapidly. When the transmission peak depth is less than 5 dB, it is difficult to extract and demodulate successfully. According to theory, better sensitivity can be obtained when the $n_{SUR}$ is less than but close to $n_{2}$. When the $n_{SUR}$ is 1.4632, it is not only difficult to demodulate but is also beyond the range required by the test. Therefore, the maximum transmission peak can only be measured when the $n_{SUR}$ is 1.4565.

### 4.2. Measurement of Refractive Index Sensitivity of Coated LPFG

TiO_2_ is widely used in the optical and biological fields due to its high refractive index, non-toxicity and high biocompatibility [8]. According to the above analysis, when the wavelength is close to 1500 nm, the refractive index of TiO_2_ is usually close to the mode conversion region (about 2.1–2.4) [28]. The $\lambda_{D}$ can be modulated effectively by coating TiO_2_ film on the surface of the LPFG.

Figure 13b shows the simulation of the response of the $\lambda_{D}$ of the LP (0.6) mode with $n_{SUR}$ when coated with different film thicknesses on ordinary single-mode LPFG samples. The sensitivity of the refractive index decreases with the increase in the coating thickness. When the film thickness is 10 nm and 15 nm, the refractive index sensitivity reaches −6832 nm/RIU and −6561.5 nm/RIU, respectively, which is about twice that of the bare LPFG in the conversion region of 1.44–1.46. Based on the above simulation analysis, the TiO_2_ film with a refractive index of 2.163 and thicknesses of 10 nm and 15 nm can significantly improve the refractive index sensing sensitivity of LPFG.

Figure 14 shows the refractive index sensitivity measurement of ordinary LPFG and Ge and B co-doped LPFG coated with 10 nm and 15 nm thickness and compared with the simulation data. The changing trend in the refractive index sensitivity of the coated LPFG obtained in the experiment is basically consistent with the simulation curve. Table 4 shows the data obtained in the experiment. With the increase in the coating thickness, the conversion area with high refractive index sensitivity moves to the low refractive index. When the Ge and B co-doped LPFG was coated with a 10 nm thickness, it obtained good refractive index sensitivity, which can provide a reference for subsequent coating sensitization.

The reason for some deviations between the experimental and simulation results may be that the optical fiber material parameters used in the simulation are not accurate enough, the coating thickness estimation is not accurate enough, and/or the experimental measurement also has some errors.

### 4.3. Measurement of Temperature Sensitivity of LPFG

Using the temperature sensing system to measure the temperature sensitivity of LPFG. When measuring, the multiple transmission spectrum peak wavelengths are measured several times after the platform temperature is stabilized, and the average value is recorded. The measured resonant wavelength shift of the LP (0.6) mode (near 1500 nm) of the bare LPFG is compared with the simulation results, as shown in Figure 15. The experimental results are in good agreement with the simulation. The fitting value of the temperature sensitivity of the ordinary single-mode LPFG is 0.109 nm/°C, and the standard error is 8.99 × 10^−4^. The fitting value of the temperature sensitivity of the Ge and B co-doped LPFG is −0.606 nm/°C, and the standard error is 0.0146. The Ge and B co-doped LPFG has significantly higher temperature sensitivity. Through calculation, the temperature sensing resolution is about 0.00165 °C, which is still not enough to meet the application of high-sensitivity ocean temperature sensing (about ±0.001 °C), and further sensitivity enhancement is still needed.

### 4.4. Measurement of Temperature Sensitivity of LPFG Packaged with UV-Curing Adhesives

The UV-curable adhesive is a polymer with a high thermoluminescence coefficient and refractive index in the mode conversion range, usually in the 10^−4^/°C range, and is an ideal packaging material. In this paper, we use two kinds of UV-curable adhesives for temperature-sensitive packaging. We measured the response of the resonant peak wavelength of the LP (0.6) mode of the sensor to the temperature. The test results are compared with the simulation results, as shown in Figure 16. The simulation data is shown in Table 2; $\alpha_{2}$ = 4.1 × 10^−7^/°C and $K_{\varepsilon }$ = 1 nm/1000 με [29]. Because it is difficult to demodulate the transmission peak when the transmission peak depth is less than 5 dB, the actual temperature measurement range is smaller than the simulation range.

The temperature sensitivity of the two packaged LPFG samples is higher than that of the unpackaged samples. In the low-temperature region, the packaging material has a larger refractive index, which is closer to the mode conversion region, so the refractive index and temperature sensing sensitivity are higher. For the simulation curves, the linearity of NOA144 is good after packaging, but the temperature sensitivity of PC452 is higher.

The experimental measurement data is shown in Table 5. In the experimental measurement, for the ordinary single-mode LPFG sensor, when the temperature is lower than 15 °C, the resonance peak disappears, which may be related to the material of the fiber itself. For the Ge and B co-doped LPFG sensor, the resonance peak remains when the temperature is as low as 5 °C.

The temperature sensitivity of the Ge and B co-doped LPFG is opposite to the influence of the thermoluminescence effect of the packaging material on its temperature sensitivity. By selecting a suitable UV-curing adhesive for packaging, we can obtain a temperature sensing sensitivity and temperature measurement range more in line with the measurement requirements of ocean temperature sensing. In addition, it is less complicated to fabricate LPFG using Ge and B co-doped fiber. In addition, packaging the sensor with PC452 has the highest temperature sensitivity, which is 3.8 nm/°C in the temperature range of 5–30 °C. The temperature sensing resolution is about 0.00026 °C when the wavelength resolution of the demodulator is 1 pm. Therefore, TiO_2_ film with a thickness of 10 nm is coated on the LPFG, and PC452 adhesive with a refractive index of 1.453 and a thermoluminescence coefficient of −3.95 × 10^−4^/°C at 25 °C after curing is used for the packaging, which is a better parameter required for coating temperature sensitization. The sensor obtained in this study is compared with other optical fiber temperature sensors, and the results are summarized in Table 6.

### 4.5. Effect of Seawater Immersion

We used deionized (DI) water and NaCl and MgSO_4_ · 7H_2_O to prepare a simulated seawater solution with a salinity of about 30% (1000 mL DI water, 3 g NaCl, 1 g MgSO_4_ · 7H_2_O). The UV-adhesive materials were solidified into sheet samples with a size of about 1.5 × 3 cm and a thickness of about 3 mm. The samples were immersed in the above-simulated seawater solution for different time, and the refractive index of the two packages was measured with an Abbe refractometer.

The refractive index of the UV adhesive changed little within seven days after soaking. This variation may be due to measurement errors caused by different ambient temperatures. Short-term soaking has little influence on the refractive index of packaging materials. However, the effect of the seawater environment and soaking time on the performance needs to be further studied.

### 4.6. Effect of Protein Adsorption

We also examined the impact of protein contaminants on the temperature sensitivity of LPFG. Fibrinogen, a common fouling agent with strong adhesion to SiO_2_ surfaces, was used to measure the temperature responses of the Ge and B co-doped LPFG (period = 400 μm; number of periods = 60) before and after protein adsorption. To assess the impact of stable adhesion, the gate area of the sensor was immersed in a 1 mg/mL fibrinogen solution for over 2 h and repeatedly washed with DI water. The temperature sensing sensitivity fitting curve of the sensor before and after protein adsorption is shown in Figure 17. Prior to protein adsorption, the temperature sensitivity test of the sensor yielded a fitting value of −0.606 nm/°C. This value decreased by approximately 33% to −0.405 nm/°C in the presence of fibrinogen, indicating that protein adhesion can significantly affect the performance of LPFG temperature sensing.

To overcome these challenges, we selected two kinds of zwitterionic polymers with a neutral electric charge and outstanding antiadhesive properties (Figure 18a): poly (HEMA-TMS)-co-PMMA (polymer B) and poly[(DMAEMA-stat-MMA)-b-(HEMA-TMS-stat-MMA)-b-(DMAEMA-stat-MMA)] (polymer ABA), to functionalize the surface of the LPFG. These two kinds of polymers fixed on the SiO_2_ prevent the adsorption of protein and bacteria. The gate area of the Ge and B co-doped sensor was immersed in a 1 mg/mL polymer B or ABA, respectively, for over 3 h and repeatedly washed with DI water. Then, the modified surface was immersed in the protein solution for more than 2 h and washed with DI water. As shown in Figure 18b, in the presence of protein, the temperature sensitivity of the sensor modified with polymer B was −0.506 nm/°C, which was about 25% higher than that of the unmodified sensor. In comparison, the temperature sensitivity of the sensor modified with polymer ABA was −0.533 nm/°C, which was about 32% higher than that of the unmodified LPFG. It is worth mentioning that, compared with the bare sensor without the protein, the sensitivity only decreased by 12% and 17.5% after being modified by polymer ABA and B, respectively, even in the presence of protein. The modification of the LPFG surface with the zwitterionic polymer improved the stability of the LPFG in the marine environment.

## 5. Conclusions

A high-sensitivity LPFG seawater temperature sensor was achieved in this study. The LPFG was carved, and a high-refractive index TiO_2_ film was coated on the surface of the LPFG to improve the refractive index sensitivity of the LPFG based on mode conversion. With the coating package, the LPFG temperature sensor with the sensitivity of 3.8 nm/°C is in the range of 5–30 °C, and the resolution is about 0.00026 °C, which meets the test requirements of ocean temperature sensing. Considering the practical application requirements of temperature sensing research in marine environments, the surface modification of the chimeric polymer coating enhances the anti-protein adhesion ability of LPFG, which can be used to improve the stability of LPFG in the marine environment. This work is useful for a high-sensitivity temperature sensor of coated LPFG in marine environment observation.

## Figures and Tables

**Figure 1 sensors-23-04768-f001:**
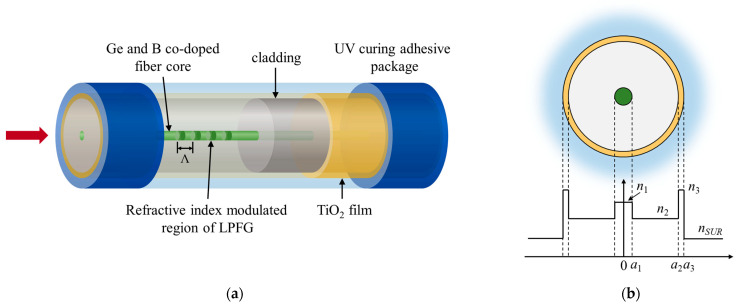
(**a**) Four-layer waveguide model; (**b**) material refractive index profile of sensor.

**Figure 2 sensors-23-04768-f002:**
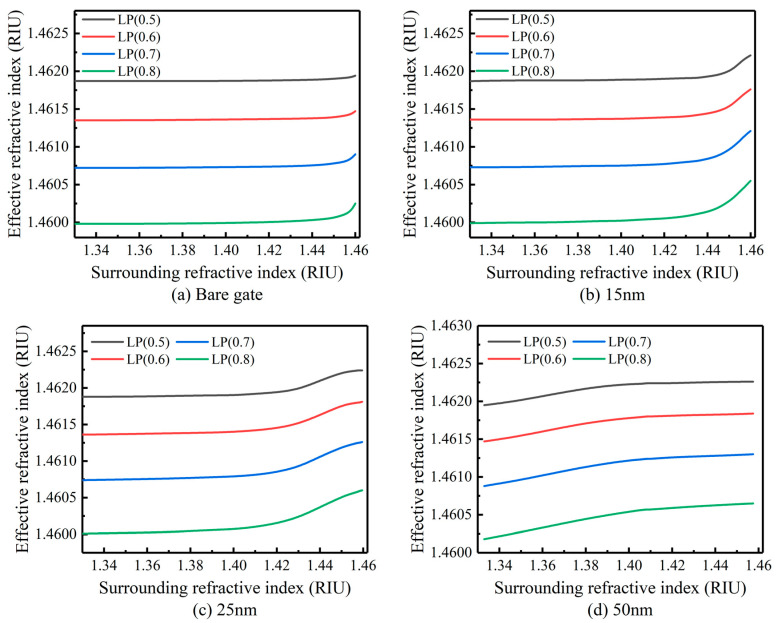
Simulation of effective refractive index of bare and coated LPFG cladding modes with SRI: (**a**) bare gate, (**b**) 15 nm, (**c**) 25 nm, (**d**) 50 nm.

**Figure 3 sensors-23-04768-f003:**
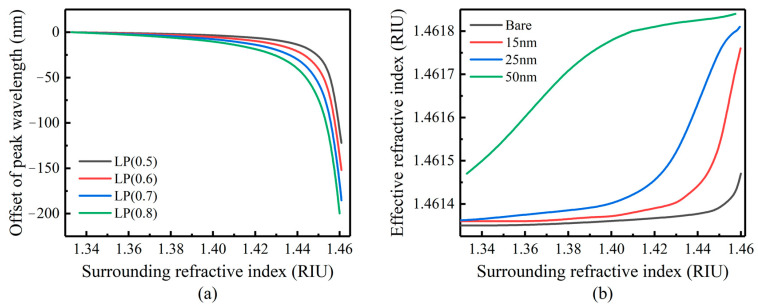
(**a**) Comparison of simulation results of refractive index sensitivity of different cladding modes when the coating thickness is 10 nm. (**b**) Simulation of the effective refractive index of LP (0.6) mode changing with SRI in bare gate and coated LPFG.

**Figure 4 sensors-23-04768-f004:**
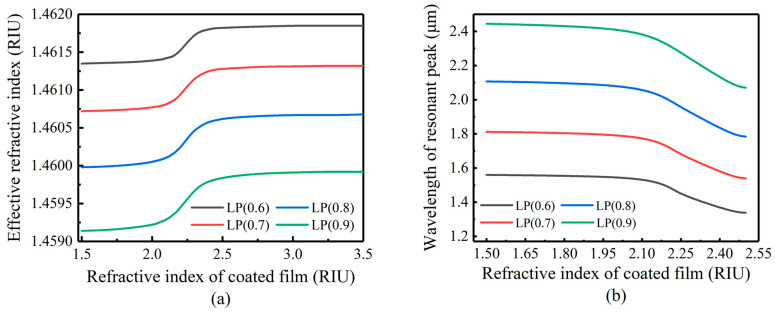
Influence of coating refractive index change on mode conversion: (**a**) cladding mode effective refractive index response; (**b**) resonant peak wavelength response.

**Figure 5 sensors-23-04768-f005:**
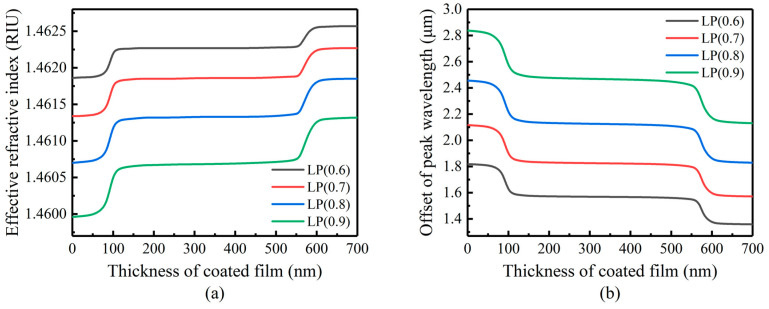
Influence of coating thickness change on mode conversion: (**a**) cladding mode effective refractive index response; (**b**) resonant peak wavelength response.

**Figure 6 sensors-23-04768-f006:**
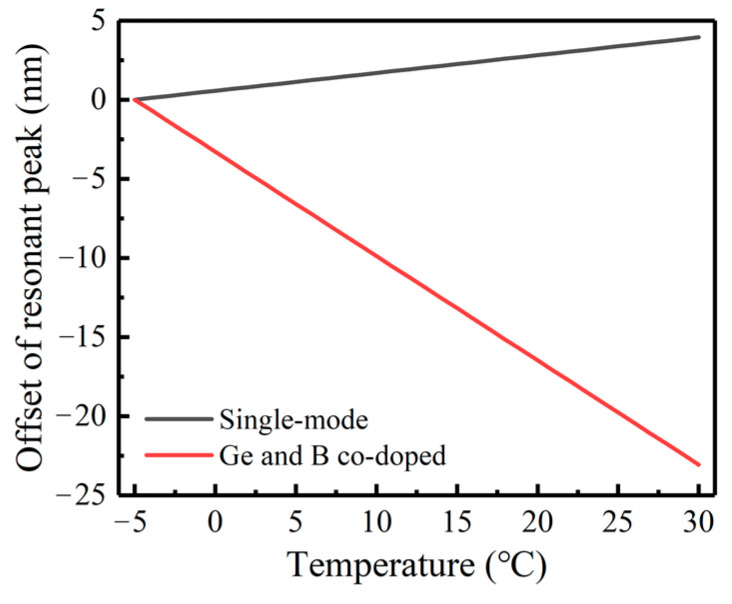
Temperature sensitivity simulation of LPFG cladding mode LP (0.6).

**Figure 7 sensors-23-04768-f007:**
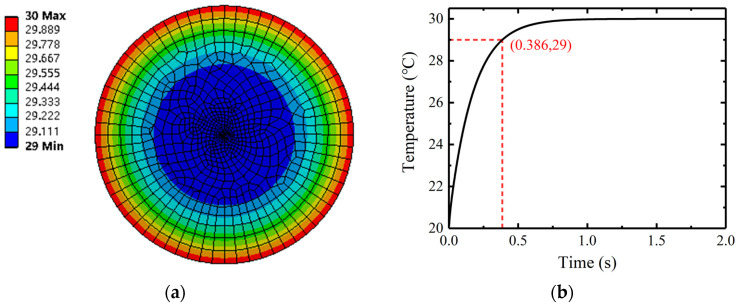
Temperature response of PCLPFG: (**a**) section temperature distribution map; (**b**) core temperature curve.

**Figure 8 sensors-23-04768-f008:**
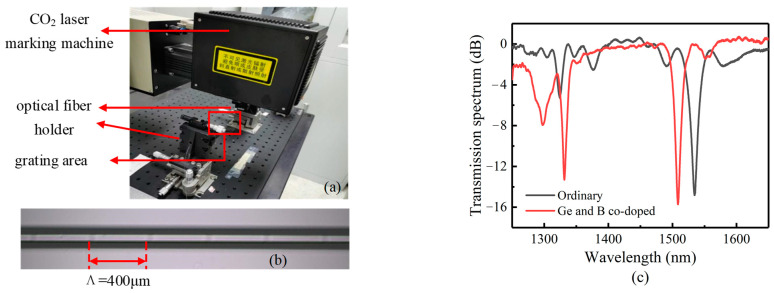
(**a**) LPFG-fabricating system of CO_2_ laser marking machine; (**b**) optical microscope photo of gate area after fabricating; (**c**) transmission spectrum of LPFG samples.

**Figure 9 sensors-23-04768-f009:**
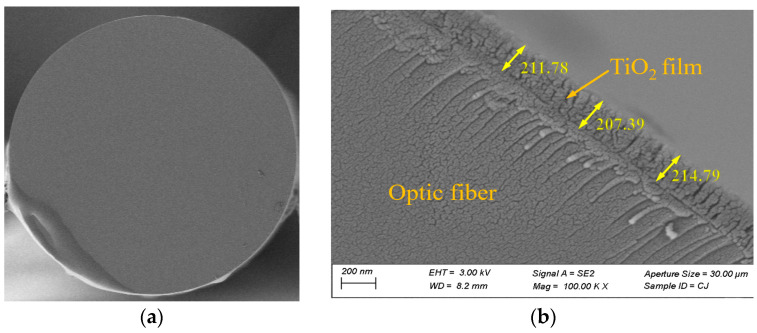
SEM diagram of cross-section of coated LPFG sample: (**a**) full section view at 1 K magnification; (**b**) partial section and coating thickness measurement at 100 K magnification.

**Figure 10 sensors-23-04768-f010:**
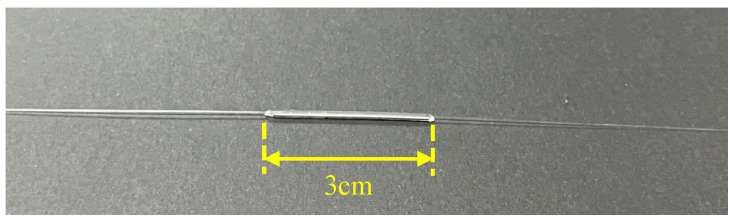
Experimental samples after packaging with UV-curing adhesive.

**Figure 11 sensors-23-04768-f011:**
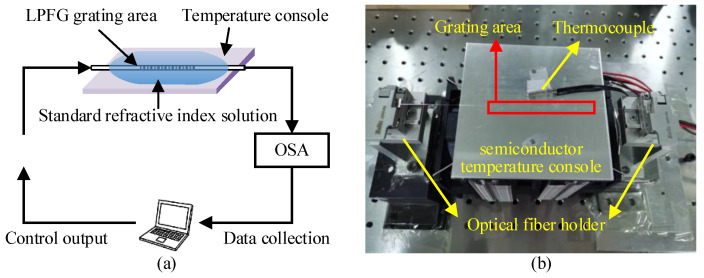
The experimental system: (**a**) schematic diagram; (**b**) physical image.

**Figure 12 sensors-23-04768-f012:**
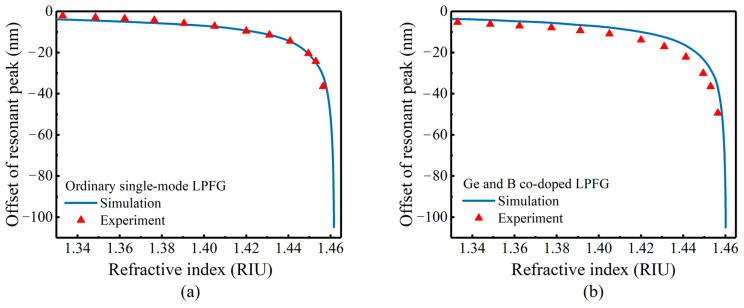
Simulation and experimental comparison of refractive index sensitivity: (**a**) ordinary single-mode LPFG, (**b**) Ge and B co-doped LPFG.

**Figure 13 sensors-23-04768-f013:**
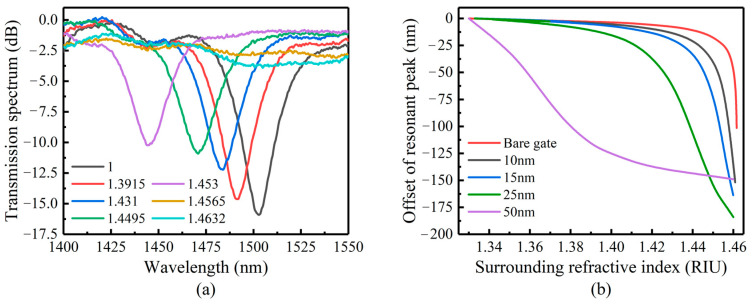
(**a**) Variation of LPFG transmission peak depth with SRI. (**b**) Simulation of refractive index sensitivity of LPFG coated with TiO_2_ films of different thicknesses.

**Figure 14 sensors-23-04768-f014:**
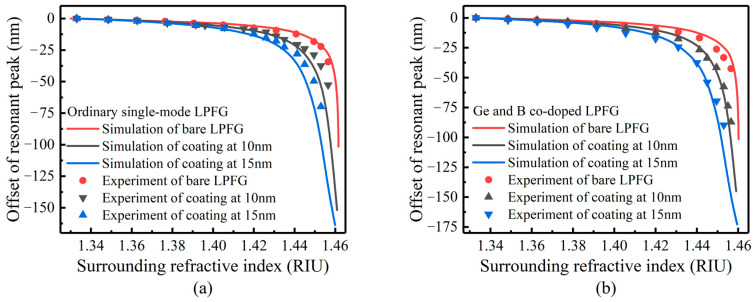
Experiment and simulation comparison of TiO_2_-coated LPFG refractive index sensitivity: (**a**) ordinary single-mode fiber, (**b**) Ge and B co-doped fiber.

**Figure 15 sensors-23-04768-f015:**
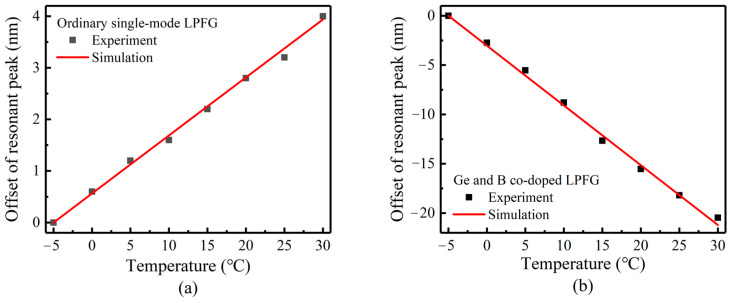
Experimental and simulation comparison of temperature sensitivity of LPFG. (**a**) ordinary single-mode gratings, (**b**) Ge and B co-doped gratings.

**Figure 16 sensors-23-04768-f016:**
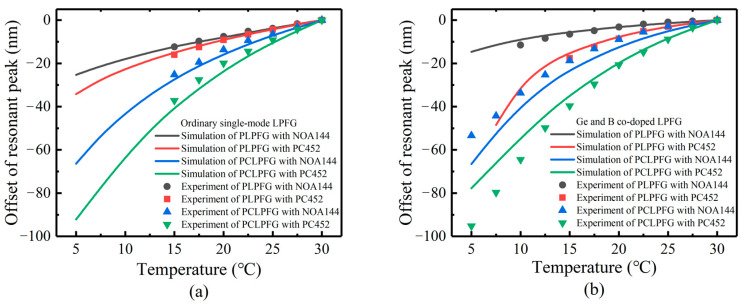
Temperature sensitivity simulation of PCLPFG: (**a**) ordinary single-mode gratings, (**b**) Ge and B co-doped gratings.

**Figure 17 sensors-23-04768-f017:**
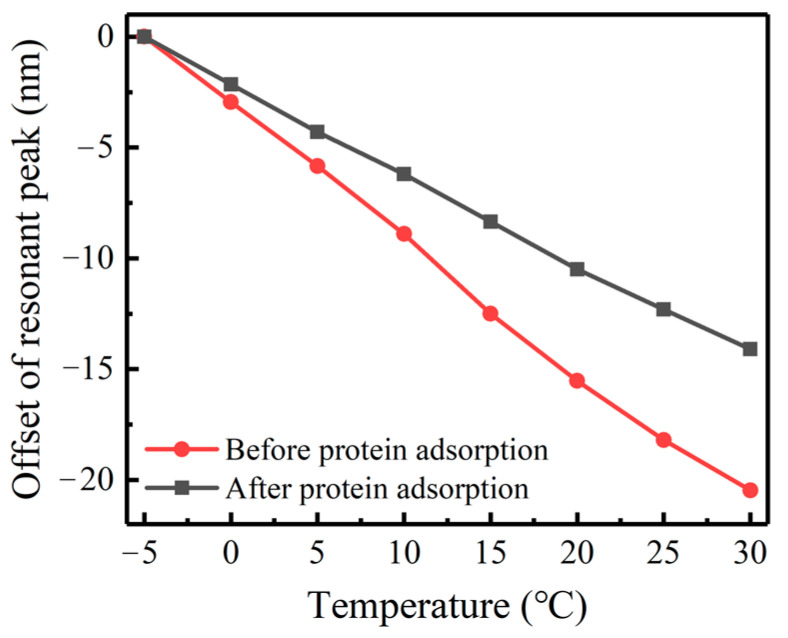
Temperature sensitivity of LPFG before and after protein adhesion.

**Figure 18 sensors-23-04768-f018:**
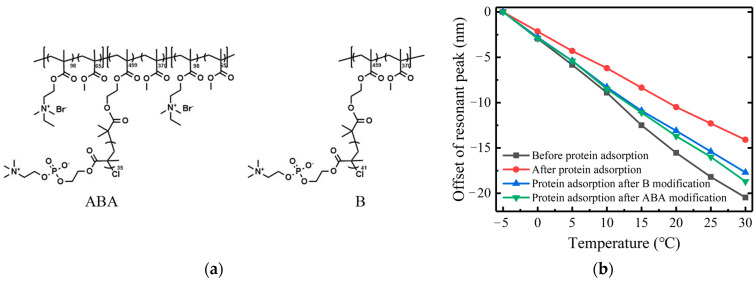
(**a**) The structure of polymer ABA and polymer B. (**b**) Temperature sensitivity test of LPFG modified with anti-protein adhesion surface.

**Table 1 sensors-23-04768-t001:** Simulation Data of LPFG Fabricated on Two Kinds of Optical Fibers.

	Core Radius	Refractive Index of Fiber Core Material	Cladding Radius	Refractive Index of Cladding Material
Ordinary single-mode optical fibers	4.25 μm	1.4681	62.5 μm	1.4628
Ge and B co-doped single-mode optical fibers	3.8 μm	1.4671	62.5 μm	1.4614

**Table 2 sensors-23-04768-t002:** Parameters of Two UV-Curable Adhesives.

Type	Refractive Index (after Curing)		Thermoluminescence Coefficient	Thermal Expansion Coefficient
	25 °C	0–30 °C		
PC452	1.4530	1.4610–1.4515	−3.2 × 10^−4^/°C	3.3 × 10^−4^/°C
NOA144	1.4490	1.4588–1.4400	−3.95 × 10^−4^/°C	3.3 × 10^−4^/°C

**Table 3 sensors-23-04768-t003:** Refractive Index Values of Standard Sample Solutions.

Number	Refractive Index	Number	Refractive Index
1	1.3330	8	1.4310
2	1.3485	9	1.4413
3	1.3624	10	1.4495
4	1.3775	11	1.4530
5	1.3915	12	1.4565
6	1.4055	13	1.4632
7	1.4200	14	1.4740

**Table 4 sensors-23-04768-t004:** Experimental data of TiO_2-_coated LPFG refractive index sensitivity.

	Coated Thickness	Surrounding Refractive Index Range	Average Sensitivity (nm/RIU)	Compare with Bare Sample
Ordinary LPFG	0	1.4413–1.4565	−1453.63	-
	10 nm	1.4413–1.4565	−2106.58	≈1.45 times
	15 nm	1.4310–1.4530	−2365.15	≈1.63 times
Ge and B co-doped LPFG	0	1.4413–1.4565	−1683.63	-
	10 nm	1.4413–1.4565	−3975.56	≈2.36 times
	15 nm	1.4310–1.4530	−2966.23	≈1.76 times

**Table 5 sensors-23-04768-t005:** Experimental Data of Samples Packaged with Two UV-Curable Adhesives.

	Temperature Range	Package Type	$\mathrm{Offset}\;\mathrm{of}\;\lambda_{D}$	Average Temperature Sensitivity (nm/°C)	Compare with Bare Sample
Ordinary LPFG	15–30 °C	NOA144	25.2 nm	1.68	≈15 times
		PC452	37.2 nm	2.48	≈23 times
Ge and B co-doped LPFG	5–30 °C	NOA144	53.4 nm	2.10	≈3.6 times
		PC452	95.2 nm	3.80	≈6.4 times

**Table 6 sensors-23-04768-t006:** Comparison of Various Optical Fiber Temperature Sensors.

Ref	Sensing Technology	Sensitivity	Temperature Range	Integration Time	Resolution
P. Zhang et al. [17]	Metal tube packaged FBG	27.6 pm/°C	0–35 °C	48.6 ms	-
L. Wang et al. [30]	FBG sensor array	26.74 pm/°C	2–35 °C	0.047 s	0.01 °C
An Jia et al. [14]	Polyamic acid-coating LPFG	1.26 nm/°C	2–35 °C	-	-
Qi Wang et al. [23]	PDMS-coated LPFG	255.4 pm/°C	20–80 °C	-	0.078 °C
B Sun et al. [4]	Fabry-Pérot interferometer	249 pm/°C	40–90 °C	-	-
F. Xia et al. [5]	Tapered MZI structure	8.33 nm/°C	24–38 °C	11 s	0.0024 °C
L. Shao et al. [6]	Sagnac ring structure	1.73 nm/°C	30–50 °C	-	0.04 °C
Yong Zhao et al. [8]	SPR	1.802 nm/°C	20–35 °C	-	0.055 °C
Q Wu et al. [7]	C-type PCF	1.054 nm/°C	0–40 °C	>1 s	0.01 °C
Y. Jiang et al. [9]	Optical microfiber coupler	5.3 nm/°C	35–45 °C	243 ms	-
Our research	New four-layer LPFG	3.8 nm/°C	5–30 °C	>1 s	0.00026 °C

## Data Availability

The study did not report any data.

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
