# Peer review of "High Sensitivity Temperature Sensing of Long-Period Fiber Grating for the Ocean"

_sensors, 2023, doi:10.3390/s23104768_

Round 1

Reviewer 1 Report (Previous Reviewer 4)

This manuscript has been improved a lot and it can be accepted.

Author Response

Thank you for your attention and comments on this manuscript.

Reviewer 2 Report (Previous Reviewer 2)

The authors answered my question very well. I agree with its publication.

Author Response

Thank you for your careful evaluation of this manuscript.

Reviewer 3 Report (New Reviewer)

In this paper, the author demonstrated a temperature sensor with high resolution for the ocean application. The paper is well organized, and the experimental results agrees well with the simulation results, therefore I recommend to publish this paper in Sensors.

1. Is it necessary to have the TiO2 film to achieve a high temperature sensitivity?

2. In the ocean temperature measurements, the sensors requires high resolution and fast response, especially for the turbulence measurements. I would like to encourage the author to provide experimental results for the sensor time constant?

Author Response

In this paper, the author demonstrated a temperature sensor with high resolution for the ocean application. The paper is well organized, and the experimental results agrees well with the simulation results, therefore I recommend to publish this paper in Sensors.

Comment:

  1. Is it necessary to have the TiO2 film to achieve a high temperature sensitivity?

Response:

Thanks for your comment, the use of TiO2 film is necessary. In this paper, refractive index sensing is carried out based on the mode coupling principle of long period fiber grating (LPFG), and then the refractive index sensitivity is converted into temperature sensitivity by the packaging of UV curing adhesive with high thermoluminescence coefficient. We coating TiO2 nano film to solve the problems of low refractive index sensitivity and limited measurement range of ordinary fiber gratings. According to the mode coupling principle, the surrounding refractive index (SRI) information is obtained by analyzing the offset of resonant peak wavelength. By sputtering TiO2 film on the surface of LPFG, the resonant peak can be modulated effectively, and the range and sensitivity of the sensor can be improved.

  1. In the ocean temperature measurements, the sensors require high resolution and fast response, especially for the turbulence measurements. I would like to encourage the author to provide experimental results for the sensor time constant?

Response:

Thanks for your comment, we agree that more study on the sensor would be useful. In this paper, we analyzed the response time of sensor through simulation. Compared with electrical sensors, the sensing speed is slow, but it can meet the needs of long-term ocean temperature sensing. In the next step of research, we will also further improve the response speed.

This manuscript is a resubmission of an earlier submission. The following is a list of the peer review reports and author responses from that submission.

Round 1

Reviewer 1 Report

In the manuscript, The author realized a new type of high-sensitivity temperature sensor by four-layer Ge and B co-doped long period fiber grating (LPFG) based on mode coupling principle. This new sensor meets the accuracy and range of general ocean temperature measurement. However, some details and discussions of the paper need to be supplemented and improved. My detailed comments are as follows:

1. p.4,line6: We established four layers model in OptiGrating software to simulate the transmission spectrum and resonance peak value of LPFG, as shown in Figure 1: Please introduce these four models.

2. p.5,line2: the sample with a film thickness of 15 nm was selected for testing, and the mode conversion range of LP(0,6) mode is 1.44-1.45, which is closer to 2n than other modes.: The mode conversion range of LP(0,5) mode is also 1.44-1.45Why not choose LP(0,5)?

3. p.9,Figure 8: Please draw a more detailed picture of the placement relationship between standard refractive index solution, LPFG grating area and temperature console. Does the depth of the LPFG grating in solution affect the sensitivity?

4. p.10,line3: “When the transmission peak depth is less than 5dB, it is difficult to extract and demodulate successfully. Therefore, the maximum transmission peak can only be measured when nSUR is 1.4565.”: In Figure 10, the transmission peak depth with nSUR of 1.4565 is also less than 5 dB. Selecting it contradicts the previous explanation.

5. p.12,line16: “Through calculation, it can be obtained that the temperature sensing accuracy is about 0.00165℃”: Please list the formula for calculating the temperature sensing accuracy.

6. p.12: Please add a picture of the actual LPFG packaged with UV curable adhesive.

7. p.14: Please introduce these two kinds of polymer coatings, type B and type ABA.

Reviewer 2 Report

The authors proposed a temperature sensor with a challenging sensitivtiy of 38 nm/℃  based on long period fiber grating. This paper can be considered to be accepted if it is properly modified.

1. The authors should use a table to compare the sensitivity and measurement resolution of international representative optic fiber temperature sensors for ocean.

2. The authors did not give detailed measurement methods and evaluation methods of temperature measurement accuracy. Temperature measurement accuracy should be replaced by temperature measurement resolution.

3. The authors should make a further analysis of the limited factors of measurement accuracy.

Author Response

请参阅附件。

Reviewer 3 Report

The paper was badly written. I think that it should highlight the new LPFG in the title. And there are many grammatical errors only in the abstract.  I even think that the author lacks the basic knowledge of fiber grating.This is absolutely not the normal level of Professor Tiegeng Liu's team. I hope the authors  rewrite the scientific paper carefully.

Author Response

对于我们稿件的语言不好,我们深表歉意,并做了一些更正。我们在手稿上工作了很长时间,反复添加和删除句子和章节显然导致可读性差。我们现在已经在语言和可读性方面进行了研究。我们真的希望流程和语言水平得到实质性的提高。特别感谢您的好评。

Reviewer 4 Report

This manuscript studied the temperature sensing of long period fiber 

grating for the ocean, the results are interesting and it can be accepted after a minor revision. The comments are as following:

1. Besides the fiber grating, other fiber optic sensing structures, including interferometer, photonic crystal fiber, and surface plasmon resonance have been also adopted by researchers, please compare their advantages and disadvantages in the Introduction.

2. Ocean temperature sensors require high sensitivity and high response speed, which must be up to illiseconds. How about the response speed in your work?

3. The pressure in the deep ocean is high. How does that affect the performance of the sensors?
